# Women's Tailored Food Delivery Platform: The Case of a Small Company in Italy

Luisa De Vita 

Department of Social Sciences and Economics, Sapienza University of Rome, 00185 Roma, Italy; luisa.devita@uniroma1.it

**Abstract:** This paper focuses on the study of "Takeve", an Italian food delivery platform that employs only women. The research conducted by interviewing both the workers and the founders of this small platform provides an opportunity to re-discuss the business models of food delivery platforms. The aim is to understand how and in what ways it is possible to initiate a participatory and multi-stakeholder model in which opportunities for fair and decent work, business-to-business cooperation, and even dialogue between companies and institutions are created. Although the case presented is circumscribed and very limited in size, it seems to be a good example to reflect on the possibility of rethinking the model of food delivery platforms in a gendered perspective.

**Keywords:** platform work; gender; sustainability; inequalities; women riders



## 1. Introduction

This paper focuses on the study of an Italian food delivery platform "Takeve" that employs only women. It is therefore a job where you deliver food or other goods and receive the order via an app. The platform was created with the goal of promoting women's employment in a growing sector, listening to the needs of both consumers and restaurants, but also small business enterprises with a focus on environmental sustainability as well. Work platforms such as food delivery platforms represent an important employment opportunity, with very broad growth prospects and no barriers to entry. However, while these jobs are easily accessible to those historically excluded from the market, such as women, the gender implications of platform work are not yet fully understood. Often the analysis of gender impacts has been omitted from research, mainly because the intensification of labour struggles in the male-dominated sectors of delivery and passenger transport in the global North has flattened the analysis of platforms to the Uber model. The increasing reference to "uberization" has obscured the study of platform labour in sectors other than delivery and passenger transport (Ticona and Mateescu 2018) and the study of how platforms operate in different national and social contexts. More importantly, it has limited a gender analysis of the different impacts that platform work has on men and women.

To deeper understand this gendered dimension, this paper analyses a food delivery platform designed to make this sector accessible and fair for women.

The research conducted by interviewing both female workers and the founders of this small platform builds an opportunity to re-discuss the business models of food delivery platforms. The aim of this paper is to understand how and in what ways it is possible to adopt a participatory and multi-stakeholder approach in food delivery platforms, creating opportunities for fair and decent work, business-to-business cooperation, and even dialogue and cooperation between companies and institutions.

This paper is structured as follows whereby the first section reviews the theoretical literature and the second part presents the method and then the results of the research. Finally, the last part discusses the results and provides some suggestions for policy makers and social parties.

## 2. Literature Review

Literature on digital labour platforms has focused on the efficiency of digital platforms in accessing labour supply. Platforms have strong growth prospects, very low barriers to entry, and they attract a wide array of earners (Hunt and Samman 2019). Platform work can be seen as a "social equalizer" by providing easy and less discriminatory access to work for several disadvantaged groups who face discrimination and inequalities based on gender, age, ethnic origins, and disabilities (Gerber 2022; Hoang et al. 2020).

However, labour platforms might contribute to reinforcing traditional gender roles and relations, increasing gender inequality. Women's entry into platform work reproduces the labour segmentation by gender (Fairwork 2023; Kampouri 2022; Piasna et al. 2022; Kasliwal 2020; Larsson and Teigland 2020). Indeed, as pointed out by Pesole et al. (2018), women's participation in the platform economy in European countries seems to be concentrated in 'feminised' tasks. The gender composition reflects the gendered nature of traditional labour markets. Remote professional and other freelance activities, including IT, are dominated by men, while women represent a large majority of on-location workers, and this category is dominated by young women providing care services (Piasna et al. 2022). This is also confirmed in the USA, where women of colour are over-represented as cleaners and home care workers, with the digital platform perpetuating the gendered and racialised subordination of disadvantaged service workers (Van Doorn 2017). Due to this horizontal segmentation, women are more exposed to experience poorer working conditions. Especially in on-location services such as delivery or cleaning services, platform work is similar to (or is a new form of) precarious work (MacDonald and Giazitzoglu 2019; Huws et al. 2018; De Stefano 2016), characterized by uncertainty, instability, and insecurity, with workers bearing most of the risks and receiving limited protections (Kalleberg and Vallas 2017; Drahokoupil and Piasna 2017). These platforms thus seem to reinforce women's placement in a secondary and more precarious labour market, and also work by increasing their economic dependence on a partner (Piasna and Drahokoupil 2017; Rubery and Piasna 2016). Significant in this regard is the issue of income. As confirmed by research on delivery platforms (Cook et al. 2021; Kasliwal 2020), women are paid less than their male counterparts.

Despite these critical issues, several researchers suggest that the platform economy can be a source of income and inclusion especially for women. Therefore, if there is a need to reduce the horizontal segregation that keeps women out of remote professional activities, especially in IT, it seems strategic to ensure fair working conditions also in secondary labour markets. These are indeed occupations that could increase the chances of women with lower educational and professional qualifications to enter the labour market. Moreover, with appropriate investment in security, training, and protection, decent working conditions could be opened to women who have been out of the labour market for a long time or who have left it because of caring responsibilities. The issue of care is another element raised in favour of platform work for women. Platform work could be a gateway to the labour market as it offers flexibility and the ability to schedule work according to personal burdens (Kohlrausch and Weber 2021). Beyond income, temporal flexibility could be an important motivator for people to pursue platform work (Schor and Vallas 2021; Berg et al. 2018), and this is particularly true for women who bear the main responsibility for unpaid care work.

Unfortunately, this flexibility is governed by algorithmic management (Wood et al. 2019). The characteristics of algorithmic management systems constrain workers' freedoms by regulating their time and activities. Platforms incentivize and push workers to work within the "right timeframe" for customer demands. Workers are asked to react immediately to customer requirements and thus to always be available especially when the service demand is greatest. These practices can also have the paradoxical effect to "lead the workers to internalize a productivist logic by promoting an ideal of hyper-meritocratic justice" (Galiere 2020, p. 368)

This model harms women more than men, as women are those who usually must juggle work with care. Indeed, if for women these jobs are likely to be their main source of income (Smith 2016), while the higher proportion of women's total working hours is unpaid domestic and care work (Howcroft and Rubery 2018), then men are more likely to have another non-platform job. Following Sun et al. (2023), more than providing flexible work, platform work is increasingly becoming 'sticky labour', subjected to a fixed schedule, algorithmic discipline, and control, plus the cultural normalisation of overwork. As Irani (2015) points out, the algorithmic management of platform work intensifies fragmentation, invisibility, and competition, creating a global labour market that can marginalise the most vulnerable groups. The flexibility offered by platforms is thus subject to trade-offs between earnings and availability in terms of working hours, increasing stress and insecurity for workers, especially women, who are overwhelmingly responsible for domestic duties.

If some research seems to suggest that, for example, the gender pay gap in ride-hailing is largely caused by women's preferences in driving speed and where they work, then it seems to be much more convincing that the risk of job loss and low pay means that women end up working long hours on platforms. (Anwar and Graham 2020). The outcome is a further discrimination in women's ability to sustain work for a long time and blurring the lines between private and professional life that may increase stress and related health problems (Smith and Leberstein 2015).

Looking more specifically at the platform of food delivery, it is necessary to highlight additional difficulties for women.

The delivery sector has been historically male dominated. Overall, the most recent data (Piasna et al. 2022) for the delivery sector shows that women represent around 40% of the workforce. As highlighted by several studies in Spain, only 13% of food delivery riders are women (Adigital 2020), while in the UK 6% of all Deliveroo riders are women (Dupont et al. 2018) and in Italy the number is 15% (Giorgiantonio and Rizzica 2018). The low numbers are at least partly because working on a bicycle is a physically demanding activity. Moreover, because women's bodies are not considered to be 'proper cycling bodies' (Aldred 2013), women are perceived to be less suitable and able in performing this work. Certainly, a relevant issue is linked to safety, related not only to the risk of harassment but also to workloads that are too high. On the first point as a minority in this sector, most women develop strategies of self-protection against hostility and sexual harassment, such as avoiding some place or times but also socialization in the workplace. For example, women avoid seeking advice from male colleagues who may have more experience with managing work or even just using apps (Popan and Anaya-Boig 2021). This has a negative impact on work continuity and earnings, and keeps women isolated and unrepresented also in demands for claiming. Furthermore, even algorithmic management to 'protect' women often opts for solutions that exclude or limit women's opportunities to work (Fairwork 2023). On the second point, the rushing nature of the work imposed by the algorithm forces riders to navigate the streets quickly while carrying bulky bags that make cycling less comfortable. (Cant 2019). If cities are often not designed to ensure proper spaces for bicyclists who are therefore less visible and more susceptible to crashes, the speed of delivery pushes them to ignore the highway code with great risk to workers' safety. Women cyclists seem to have a higher exposure to road unsafety. In Aldred and Crosweller's (2015) study, women reported more near misses per hour and per unit of distance, with 50% more close passes per unit of distance than men. Moreover, since access to toilets is a basic and frequent need for women's health, sometimes in addition to the difficulty of finding available hygienic services, the request to use the toilet made to restaurants may pose additional safety risks to riders.

The combination of these issues requires the need to rethink the organizational model of platforms while avoiding a gender-neutral perspective. Certainly, in improving the working conditions of women in the platform economy, an important role is played by welfare systems. Welfare policies can aggravate or mitigate the insecurities and risks already discussed (Gerber 2022). However, the way in which platforms are managed, the use of

algorithms to modulate workloads, and the possibility to introduce a gender-sensitive criteria in the organizing of labour requires a different organizational model. One possible alternative is cooperative platforms such as worker-owned and worker-governed platforms (Schneider 2018; Benkler 2016; Scholz 2016).

These are platforms in which workers have more control over their income and the surplus value can be redistributed to benefit producers, support workers' welfare, and invest in workers' training and education (Hunt and Samman 2019). The distinctive principles of these cooperative platforms are equal work relation, collective ownership, democratic decision-making process, open-sourced software, and transparent algorithms and data collection (Peticca-Harris et al. 2020; Vallas and Schor 2020). Thus, it is expected that this platform should be more sensitive to gender issues, far from solely masculine work cultures, and more attentive to the unequal gendered division of domestic labour (Van Doorn and Badger 2020; Van Doorn 2017). Although these cooperatives are still at a very early stage, they seem to represent an alternative way of organizing work and to be moving the platform world towards more inclusive systems. There is already some research on these cooperatives, such as "Mensakas" in Barcelona, co-founded by women with a clear feminist perspective, paying women 5% higher wages to compensate the gender pay-gap in the sector. Another interesting case is the Hong Kong Women Workers' Association (HKWWA) that through prioritizing women workers' health provides child-care services for women workers, thus helping especially migrant women to better manage care burdens.

These cooperative platforms seek to address some of the critical issues of platform work by securing fairer conditions for their workers, but they also raise a number of questions, particularly about whether they can really provide an alternative to the monopoly power of large providers in, for example, food delivery. From this point of view, an important role is also played by the institutional context, welfare, and labour policies in fostering and supporting different and more human friendly business models also in a logic of local development. This research will analyse an Italian platform that, by adopting a business model specifically aimed at and designed for women in the food delivery sector, represents a particularly interesting case study, especially in relation to the partial regulation that has affected the delivery sector in Italy but that still seems not to have achieved the expected results.

### 3. Materials and Methods

This research focuses on a case study of the female driver platform 'Takeve'. The decision to use this case study was based on the need to understand in depth the organizational model, management style, and culture of this new delivery platform model. Case study is a qualitative methodology that allows for the study of a contemporary phenomenon in its real context. It makes use of multiple data sources including archival data, survey results, interviews, focus groups, ethnographic studies, participant observation, videos, photographs, etc.

Its purpose is not to generalize the case under consideration, but rather to understand it in its specificity, uniqueness, complexity and in its specific social and economic context (Mabry 2008; Stake 2005). In the case of a case study involving an organization, as in the present research, the advantage is that it is possible to examine all the different activities that are carried out within the organization and which, although part of the daily routine, are not the subject of reflection or analysis. The case chosen represents what can be called a "revealing case" here of a new business model in the platform economy and one that has not yet been sufficiently investigated.

Operationally, the first phase was an on-desk analysis to find all the information materials about the company. Website pages were used, but all institutional and non-institutional sources, online and in print, that in various ways mentioned the company were also collected. In the second stage, in-depth interviews were conducted with the two founders of the company.

This type of interview was chosen on the grounds that it is more suited to exploring and reconstructing the various meanings that actors grant to the context in which they operate. Indeed, our aim was to delve into the interviewees' points of view by soliciting in-depth responses. Furthermore, the "[...] flexible and non-standardized interrogation structure" (Corbetta 1999, p. 405) of this type of interview offers researchers considerable flexibility (Arksey and Knight 1999; Patton 2002) in that the interview process can be tailored to interviewees. Thanks to this flexibility, we were able to explore the topic in depth by adapting our interview questions.

The interviews were repeated on several times to track the transitions the company was putting in place. Indeed, the current organizational model is the result of a series of adjustments made over time, for example, in the provision of security equipment or in the management of customer relationships.

Instead, the third phase involved employees from the two locations where the company operates, Milan and Rome. Four focus groups were carried out that allowed for an in-depth investigation, thanks to the participants' interactions, not only of their experience with the Takeve platform but also of their own experience as riders on other platforms. Indeed, the focus group makes it not only possible to reconstruct the attributions of meaning (Tourangeau and Rasinski 1988) that different participants give to the various issues under investigation, but also through a process of sharing and comparing (Morgan 2002) to clarify individual positions and compare them with those of others.

The focus group also enables information gathering during the discussion (Tourangeau and Rasinski 1988) with an information amplification effect.

The opinions gathered will be greater and different from the sum of the opinions that would be obtained by interviewing these people individually (Scardovelli 1997). In fact, the focus group encourages the expression of a variety of positions and definitions of the same situation, activating the recollection of forgotten details and aspects not personally considered; this dynamic does not occur in a two-way interview where the interaction, while present, is between asymmetrical actors. The choice of the focus group is therefore well suited to obtain novel responses, unexpected opinions, and unanticipated aspects, thus stimulating the researchers' interpretive imagination (Bertrand et al. 1992; Dawson et al. 1993).

## 4. Results

Takeve is a very small delivery company launched in 2021 with solely female riders. The company, founded by two female entrepreneurs active in the food service and marketing sectors, has a total of 12 employees and is based in Rome and Milan, although the two cities are not entirely served. Delivery services, indeed, are available only in a few neighbourhoods.

### 4.1. The Demand Side

Interviews with the two founders immediately revealed their intention to enter a high-growth sector by promoting female employment in an industry considered strategic for women, especially in terms of flexibility.

The decision to establish this business was based on a preliminary examination of the industry, which gave the two female entrepreneurs a thorough understanding of its criticality, particularly in regard to gender-based discrimination. Before starting the company, one of the two founders worked as a rider for a short time. This helped her understand how to create a safe and good workplace for her employees. Interviews with the two founders revealed an organizational model based on a managerial philosophy that is redistribution-oriented, multistakeholder, and sustainable for workers, customers, and suppliers.

The first distinguishing element is the guarantee of fair conditions of work. All riders have a regular contract, receive a wage not tied to the number of deliveries, and are covered by insurance, which is additional to the social protections and safeguards provided by the

contract. The organization of work is designed to ensure the safe performance of work. In fact, there are no indications/impositions regarding the speed of deliveries, and the service is stopped in case of severe weather conditions.

> *Here in Rome, we have on several occasions sent riders by taxi to pick up their orders and the weather was very bad. Yes, we want to promote environmental sustainability with electric vehicles, but we also want to protect riders without putting them at risk. In extreme weather, the platform stops and customers have to wait until the weather improves. Or, in desperate times, a taxi may be called. Here, too, we need to be aware of the customers who order from us, the protection of the riders, and the safety of the passengers.* [Founder 1]

Also on the safety side, riders are provided with a whole series of facilities both for safety related to the type of work done on the street, with traffic, etc., and with respect to the risk of violence. On the first point, riders are provided with ad hoc equipment. From this point of view, the partnership with a famous sportswear brand to supply shoes designed to provide maximum comfort but also greater safety in terms of grip on the bike pedals is significant. To reduce the risk of violence, all riders are equipped with GPS tracking and a wristband with an audible alarm that can be activated in the event of danger. The platform has always used a contactless delivery mode to minimize contact between the worker and the customer and to avoid potential problems. In addition, a Takeve point (currently only available in Rome) has been opened to ensure greater safety for riders. This is a place where riders can rest, go to the toilet and, above all, not be forced to wait in the street between one delivery and the next.

The second element is the adoption of a model that is not exclusively based on profitability, but on redistributive mechanisms. As already mentioned, the riders are fairly paid and are employees. The possibility of keeping the company profitable by offering quality services is based both on the ability to intercept a series of public funds (national or European) and, above all, on the creation of collaborations with other companies that can provide Takeve with a series of services with important facilities. This is the case with the clothing, for example. The electric scooters used by the riders are also the result of a collaboration between the platform and a sustainable mobility company.

> *We can't buy everything, we're small, we don't have the resources, but we share our philosophy with other companies and then it becomes possible to provide a quality service at a lower cost.* [Founder 2]

The founders also turned down several offers to buy or take over (from other delivery platforms), so as not to denature their model and bend it solely to the logic of market competitiveness. Even the company's headquarters were symbolically placed in Rome rather than Milan. This is because Rome is a less mature context for sustainability issues, especially in terms of suppliers (restaurants) and consumers. The cost of the service is higher for restaurants that choose Takeve's delivery. Fair pay for the workers, the use of electric vehicles, and the provision of a range of equipment for the riders represent significant organizational costs, especially for a small business. Those using the service, both consumers and suppliers, must therefore share the platform's philosophy and choose to pay more. From this point of view, Milan is a more mature and sensitive context for these issues, while in Rome, perhaps due to a more fragile socio-economic context, the choice is still more about convenience. The aim is to raise awareness by creating a new culture around work, its value, and the people who do it.

The third element is therefore sustainability. Obviously, social sustainability is ensured by the choice to dedicate only to women and to guarantee fair conditions. Environmental sustainability, on the other hand, is primarily achieved by using all-electric vehicles. When we first interviewed the founders, the platform was in a period of suspension, both because all the workers were involved in a training course on road safety and hygiene standards for food processing, and because the purchase of the electric scooters, which were to be loaned to the workers, was being finalized. Secondly, the platform promotes a zero waste campaign

to avoid wasting unsold food. The service offers partner restaurants the opportunity to create customized last-minute offers that combine cost savings and timeliness of service. Those who choose Takeve also automatically donate part of the proceeds to non-profit organizations active in projects against violence towards women.

The fourth distinctive element is related to the development of the territory by favouring proximity shopping. The platform not only delivers groceries, but also provides a home delivery service for other types of goods through a series of agreements and partnerships with small shops in the neighbourhoods served by the platform. This service, although still in fieri, gives small neighbourhood shops the opportunity to expand their sales network or even consolidate their relationship with their customers by offering a new service. It also gives riders more flexibility by giving them a wider choice of times to work, which do not necessarily coincide with the busiest times for restaurants.

The final distinctive element, also mentioned above, is the building of a multi-stakeholder network, both with other companies for partnerships and collaborations in the provision of services and equipment and with different institutional actors. Certainly, the dialogue between Takeve and applicable institutions has been fostered by a series of institutional recognitions that have rewarded the organisational model and the mission. In fact, one of the founders was recognised by the President of the Italian Republic as an innovative entrepreneur capable of introducing new inclusive business models. The focus on safety, on the other hand, has enabled the company to establish relationships with the Labour Inspectorate and the National Institute for Industrial Accidents (Inail) to initiate safety training projects for its employees. In addition, Takeve has been able to collaborate in the design of training interventions aimed at the entire sector, which as is well known faces several critical issues related to illnesses and the risk of accidents. Institutional recognition has also been accompanied by a great deal of interest from the media, who have given the project a great deal of publicity and, to a certain extent, supported and encouraged collaboration with other companies in a common cause. Unfortunately, however, as the entrepreneurs themselves noted, institutional support at present is mainly in the form of endorsement but has not resulted in the actual provision of dedicated resources.

> *I have shaken so many hands, received so many promises, and now where are they all?*
> [Founder 1]

What has been most fruitful, therefore, has been the opportunity to participate in public policy arenas and to build relationships with a range of actors, such as employers' organisations, who share, at least in part, different visions of business and support an inclusive environment that is more positive towards new entrepreneurial actors.

*4.2. The Supply Side*

On the worker side, the focus groups confirmed the adoption of an organisational model very different from that of the food delivery platforms. The respondents are almost equally divided between very young girls (between 23 and 26 years old) who took up the job of rider during their university studies, and older women between 33 and 45 years old with children who, by contrast, have had several work experiences and have found in this job the possibility of combining paid work with family responsibilities. Only one respondent is of foreign origin.

One of the most frequently mentioned elements is the lack of pressure in Tekeve to place as many orders as possible in the shortest possible time, as well as the possibility offered by the platform to suspend the service in the event of critical weather conditions. As the quotes below show, the dimensions of safety and freedom from pressure are very important to these women workers.

> *In XX I had to make a delivery in less than 10 min. Riding a bike is dangerous. What I really like about Takeve is the timing, i.e., no such tight deadlines within which to deliver.*
> [Riders 1]

*It is very important not to have the pressure of delivery time. And I think that is something that is absolutely unfair and also puts you in danger, in short.* [Riders 3]

*For example, if it's raining or sunny and you're really bothered by something, they'll say, 'That's it, we're going to suspend the app until the weather clears and you can come back safely and quietly'. It wasn't like that in XX. You had to deliver regardless of the weather.* [Riders 1]

On the other delivery platforms, the times imposed by the algorithmic management are certainly discriminatory for women. As reported below, the very low presence of women in this sector is also due to the difficulty of meeting the thresholds imposed by the platform and to the high risk of accidents.

*Because it had a very specific timeframe. So if I did, I don't know, 30 deliveries, they would average me out. I was always averaging 6 to 7 min. The men, on the other hand, did less than 5 min, so there was a difference of two or three minutes. I did everything I could to meet the delivery time. . .* [Riders 5]

Especially for those who have been involved in accidents in the past because they had to meet deadlines, safety becomes a key factor in the decision to continue working. This is confirmed by the testimony of a rider who, although she had found a job better suited to her training, decided to continue working with Takeve precisely because she could do a job she liked in complete safety. Surprisingly, the focus group revealed that one of the most appreciated elements is not the contract that guarantees a fixed salary, social protection, etc., but the fact that they are at the centre of the company's interests, which ensures the well-being of its employees even at the expense of economic loss. Although the focus groups did not reveal any testimonies of women who had suffered harassment or found themselves in dangerous situations, the presence of a series of devices (the GPS and the audio bracelet) and the contactless mode were perceived by the riders as additional elements aimed at improving their working conditions. Even the platform's shutdowns, when riders were asked to stop work to attend training courses and give the company time to refurbish its vehicles, were welcomed by the workers. They fully share the company's philosophy and are willing to give up work in order to maintain the platform's high standards for both new recruits and end users.

The focus on safety is not only seen as a protective element, but also as an enhancement mechanism. These women like being riders, even if it is occasional or part-time work. They like being outdoors, having control over their own time, and having a safe job where they feel valued. Thus, security issues are closely linked to valorisation:

*Knowing that I was working for a company and that I had support, that there were people there to protect my work, that was a lot. Just one evening, for example, it was raining a lot and it was Evelyn herself who told me 'stay in this covered area, if there are orders, let's see if they can be reached, otherwise let them know that you can't go because of the rain', because it was really dangerous. I worked mainly in the autumn, so the weather was often like that. And this is clearly not the case in other delivery situations where people put themselves in danger to deliver their orders.* [Riders 4]

*The riders are waiting in the middle of the road. At least that is what I see. So I think you have to protect the riders in this aspect, have some kind of meeting point or I don't know what. Here in Rome we have the Takeve point, a central place to have a reference point. You need a reference point. . . it's fundamental, especially for a woman.* [Riders 5]

Feeling protected, not only from a contractual point of view but also in terms of physical safety, increases satisfaction and, above all, gives a different status to what is often seen as a low value job. The need to change the image associated with riders, by giving dignity to food delivery work and bringing suppliers and consumers closer to a better understanding of the problems and needs of delivery workers, is a distinctive feature of their business and a source of pride for them. The idea of working for a sustainable company, both in terms of working conditions and in terms of the environment and

neighbourhood networks, is one of the elements that the riders believe should be given more emphasis in the media.

The women workers see Takeve as an attempt to give dignity to this work. During the focus groups, many said that they started with a certain amount of fear, not only because of the risks involved, but also because of the social stigma attached to riders. As we can see from the quotes below, the job of a rider is often perceived as a second-class job, devaluing those who do it, whereas Takeve's action seems to go in the opposite direction:

> *The premise of Takeve was really to say 'let's turn this industry around, in every sense, with all the protections', so that was what made me say 'OK then I want to get involved because it can make a difference to the whole industry'. Within Takeve we have all the protections, a fixed salary, fixed weekly hours, safety equipment, and all sorts of regulations. Shortly after I joined Takeve, I saw that other deliveries had also updated their safety standards and things like that. But from what I see with my own eyes, I don't really see that the situation has changed much. For some things I see that there are improvements, and that has made me change my mind about the industry. I see it much more as a normal job, I don't know how to say, not exploitative.* [Riders 2]

> *. . .this is also an ethical supply, the customers we serve agree with us, that is, they agree with the idea that Takeve brings forward, and so they are also good and tolerant if, for example, there is a little delay or if there has been some misunderstanding about the order. As there are only a few of us, there are also loyal customers that we know, so there is also a relationship, however short, however fast, that is nice.* [Riders 4]

> *What I would like to change, but I am sure that at least in Takeve they are working on it, is to spread the word. . . that there is an ethical supply and that it is possible. That there is a supply that not only offers work, but also offers training to those who have to work in this sector, which at the moment is a bit of a wilderness, not regulated by anything, so that those who are more reckless earn more, risk more, but earn more, and instead those who take care of their skin earn less. The fact that it is possible to change all this means that someone is already doing it.* [Riders 5]

> *As far as my experience was concerned, everything was going in the right direction, and that's what convinced me. I hope this will also change the general perception of this work. Even from one delivery that works in this way.* [Riders 1]

As expected, the small size of the enterprise facilitates the possibility of establishing a direct and permanent relationship between the founders and the workers. This is probably a practice that is destined to change, and in this sense Takeve's features also seem to be designed to favour self-help mechanisms among female workers, thus limiting the intervention of founders.

## 5. Discussion and Conclusions

This study has several limitations. Firstly, it is a very small organisation operating only in Italy, which significantly limits the possibility of generalising the research findings to other contexts. Secondly, the research only analyses one platform, whereas it may be useful for future research to compare the Takeve model with other platforms to better analyse gender impacts in relation to different ways of organising work. Finally, the company was launched in November 2021, making it difficult to assess how and whether Takeve's organisational model can be a true alternative to traditional delivery platforms. Despite these limitations, the case presented seems to be a good example to reflect on the possibility of rethinking the model of food delivery platforms.

The model is designed in a non-gender-neutral logic that takes into account women's needs. The proposed business model is based on the recognition that the nature of the work carried out on the street, in contact with people who are always different, presents much higher risks for women than for men. Similarly, the different bodies of men and women require different working rhythms. This is not to say that working conditions bordering on exploitation, typical of food delivery platforms, are good for men but not for women, but it

does underline the need to ensure that while certain conditions are useful for everyone, for women they are crucial to the decision to continue working.

The fact that it is a platform founded and managed by two women also seems to make a difference. As several studies have shown, although female entrepreneurship is characterised by more limited performance and smaller size (Jennings and Brush 2013; McAdam et al. 2019), female entrepreneurs are often able to take advantage of the opportunities offered by digital technologies to create social innovation (Tracey and Stott 2017). If following Edwards-Schachter and Wallace (2017), we define digital social innovation as the use of digital technologies to create, implement, and deliver new ideas, products, services, or models to address social problems, the actions of the female entrepreneurs of Tekeve provide interesting insights. First, we can also look at the labour platform through a gendered lens (Lindberg et al. 2016), reversing a narrative that tends to see women only in terms of underrepresentation. Second, it would provide an analysis of the growing number of women entrepreneurs who 'find innovative solutions to social problems that are not adequately addressed by the local system' (Bacq and Janssen 2011, p. 382). As we see in our case, fair working conditions, attention to the environment, support for causes related to gender violence, etc., combined with a leadership style that is also attentive to relational aspects, represent a gendered approach to the supposed neutrality of the food delivery platform.

To summarise, the two most interesting elements of the case presented are the adaptation of food delivery to the needs of female workers and the choice of a management model that is not based on speed and maximization. On the first point, while it seems to be true that the flexibility offered by this type of work can be an added value, especially for women with care burdens (Schor and Vallas 2021; Berg et al. 2018), being a woman requires the adoption of management that takes differences into account (Fairwork 2023; Anwar and Graham 2020). As we have seen, both in the founders' thinking and in the words of the riders, being a woman increases the risks and requires the adoption of non-gender-neutral performance criteria and the provision of different services to meet the needs of men and women.

On the second point, the absence of algorithmic management eliminates many of the critical issues that make platform work particularly burdensome (Sun et al. 2023). The presence of an employment contract is certainly one of the elements that guarantees women workers the possibility of working without necessarily having to make as many deliveries as possible and is in line with many analyses that have underlined that it is precisely the autonomous nature of the work that represents one of the greatest elements of vulnerability (Defossez 2022). Indeed, the pay linked to the number of deliveries forces them to accept all orders and to work long shifts, with obvious gender discrimination (Sun et al. 2023; Irani 2015). In Takeve, the existence of a contract and a series of practices aimed primarily at ensuring worker's safety eliminate many of the elements of insecurity and risk associated with this profession. This management is possible mainly because of the very small size of the company. But it certainly seems to open market space for new entrepreneurial actors who can help spread a different kind of work culture in the platform economy.

Thus, the element of real innovation seems to lie in the attempt to build a more inclusive and multi-stakeholder model of the platform economy. It is precisely their small size that does not allow them to survive and provide a quality service without developing networks and partnerships with other companies.

The challenge is to build partnerships with companies that are sensitive to social and environmental sustainability issues, and to create collaborations based less on cost-cutting and profit maximisation and more on shared values and the development of new business models. Linked to this is an attempt to raise awareness of the problems faced by all food delivery workers. From this point of view, the focus on women serves to highlight a number of difficulties and the stigmatisation of workers in the sector. As the focus groups showed, riders' problems are not only related to low wages, but also to the constant devaluation of these workers, which also leads to a lack of political interest. Even if thanks to a series of

strikes some improvements have been made, for example, in terms of pay (in Italy, Just Eat workers are now employees) or the introduction of a few bonuses for deliveries in adverse weather conditions, there is no serious debate about the risks of this work and the need to regulate the sector by also providing training and special services.

However, there are several critical issues about institutional support for organisations such as Takeve. It is obvious that such a small organisation is extremely dependent on external resources and on the possibility of accessing public funding to continue operating. As the case presented shows, the delivery service was interrupted to allow the platform to redefine its contract for the use of electric vehicles. These breaks, if sustained during the company's expansion phases, can affect the platform's resilience in the long run.

Clearly, the role of the public actor is strategic in its ability to support more inclusive business models. Now it depends on the two entrepreneurs' ability to build a whole series of networks on their own. Instead, incentivizing the use of services provided by companies with a clear commitment to sustainability could encourage not only a new way of doing business, but also a new way of consuming. As we have seen, choosing Takeve means that both the customer and the consumer are willing to pay more for the same service that other food delivery companies offer at a much lower price. The real possibility of choosing which delivery to use is thus tied to the individual resources of the customer or consumer and could lead to further segmentation of the market on both the worker and consumer/supplier sides.

The possibility of a real competitive model for the large delivery platforms also depends on the need to make the services offered by these companies more affordable and/or, in any case, to enable them to really represent an alternative. So far, this is clearly a business model that, while promising, has had very limited impact. To sustain this new model, the social parties' role is again strategic.

The dialogue so far seems to be mainly within the world of employers' organisations, which are starting to push for reward criteria that recognise companies' commitments to inclusion and sustainability. Trade union action, on the other hand, is still too focused on wage dynamics and often pays little attention to gender impacts. As industrial relations scholars have pointed out, trade unions tend to adopt a gender-blind lens and thus fail to problematise the fact that their activities reflect standards based on specifically masculine norms. This unexamined bias is clearly manifested in gender-blind mechanisms that simply accept the gender-neutrality embedded in, for example, collective agreements (Dickens 2000) and organisational practices (Acker 2006) such as algorithmic management. Despite these limitations, if there has been a minimal change in the working conditions of riders in Italy, it has been due to the joint efforts of workers and trade unions (Tassinari and Maccarrone 2020). This suggests not only that traditional struggles are still useful in the new digital economy (Fernàndez and Barreiro 2020), but also that unions can also play a key role by extending their influence beyond collective bargaining to a broader regulation of the labour market and its actors in a gender-sensitive logic.

In conclusion, the new model proposed by Takeve may not take us very far, but at least they present an alternative vision that is worthy of support, especially from social dialogue actors.

**Funding:** This research received no external funding.

**Institutional Review Board Statement:** Not applicable.

**Informed Consent Statement:** Informed consent was obtained from all subjects involved in the study.

**Data Availability Statement:** Data is unavailable due to privacy.

**Conflicts of Interest:** The authors declare no conflict of interest.

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
