# Peer review of "Women’s Tailored Food Delivery Platform: The Case of a Small Company in Italy"

_socsci, doi:10.3390/socsci12090512_

Round 1

Reviewer 1 Report

This research examines the platform of food delivery of a small, female-owned and operated company and its challenges for women workers. Although not generalizable, the study is novel in its approach to the topic of orginsational platforms and serves as an example to future companies regardless of type. The researchers point out a main problem, discrimination against women, regarding traditional platforms. Additionally, it points to gender neutrality of rider work as unsubstantiated. It's serves as a suitable study for future research to extend. 

Suggestions

1. Include a 'Definition of Terms' section since there are several terms and concepts (e.g. delivery platforms, models (lines 21-22). This could be placed between 'Introduction' and 'Literature Review' sections. 

2. Lines 25-26 - Language is vague: "...recently..." and "For a long time..."

3. Line 34 - Appropriate place to begin "Literature Review" section

4. Lines 98-100 - Is there more recent data (2022-23)?

5. Line 139 - Cooperative Platforms aren't "new." Is this in the context of food delivery programs, specifically? What does "new forms of organisation" (Line 149) mean?

6. Line 170 - Insert space and parenthesis. 

7. Lines 189-190 - What is meant by, "...transitions the company was putting in place." What transitions? 

8. How many participants in each focus group? 

9. Lines 211-214 would be more appropriately placed beginning on Line 180. 

10. Providing background about participants (e.g. demographics, socio-economics) would provide more context, which would strengthen the study.

11. Include limitations of the study/research.  

Author Response

Thank you for giving me the opportunity to submit a rewrite version of manuscript Women's tailored food delivery platform. The case of a small company in Italy.

We appreciate the time and effort that you and the reviewers dedicated to providing feedback on manuscript and I’m grateful for the insightful comments on and valuable improvements to my paper.

I have incorporated most of the suggestions made by the reviewers (Red changes in the uploaded file). Please see below, for a point-by-point response to the reviewers’ comments and concerns.

Q: Include a 'Definition of Terms' section since there are several terms and concepts (e.g. delivery platforms, models (lines 21-22). This could be placed between 'Introduction' and 'Literature Review' sections.

R: Thank you for this suggestion, I have clarified and specified it better.

Q: Lines 25-26 - Language is vague: "...recently..." and "For a long time..."

R: I have modified the file

Q: Line 34 - Appropriate place to begin "Literature Review" section

R: In the first version I had followed the journal's instructions to have a single paragraph between the introduction and the theoretical framework, now I have divided the two parts and rewritten the introduction. 

Q: Lines 98-100 - Is there more recent data (2022-23)?

R: I have included recent data for the delivery platform sector as a whole, but unfortunately I was unable to find disaggregated data for each country.

Q: Line 139 - Cooperative Platforms aren't "new." Is this in the context of food delivery programs, specifically? What does "new forms of organisation" (Line 149) mean?

R: I have tried to make it clearer

Q: Line 170 - Insert space and parenthesis.

R: done

Q: Lines 189-190 - What is meant by, "...transitions the company was putting in place." What transitions?

R: I have tried to make it clearer

Q: Lines 211-214 would be more appropriately placed beginning on Line 180.

R: Thanks for the suggestion, but I preferred not to change, it seemed more logical.

Q: Providing background about participants (e.g. demographics, socio-economics) would provide more context, which would strengthen the study.

R: I have included the background data, but have not gone into detail because, due to the small number of employees, providing more detail runs the risk of making respondents easily identifiable.

Q: Include limitations of the study/research.

R: I have included the limitations of the study in the conclusions

Reviewer 2 Report

Dear Authors,

I wish to thank you for the possibility of reading and reviewing your paper. You undertook a significant problem of tailoring the workplace for women, the problem which is worth discussing and publishing. However, I found some issues requiring the improvement to make your paper more clear:

1.       The focus of the paper should be clear from the starting point. You refer to ‘food delivery platform’ in the title; to ‘platform work’ in the first sentence of the abstract; and to ‘women riders’ in the second sentence. I felt lost with your main topic after those first lines. Please rephrase these parts to be strict with your main issue from the beginning.

2.       Abstract requires to be re-written. Look at your first and fourth lines of the abstract. In the first, you describe the aim (let me quote): “This paper aims to analyze the gender impact of platform work”, then you make some justification, and then again in the 4th line you come back to your aim, however, you write on different aim than in the first line: ”The aim is to understand how and in what ways (…)”. You need to be strict with describing your aim and with keeping the structure of an abstract: justification, aim(s), method, results, novelty.

3.       The introduction needs to be re-structured. Typically, an introduction presents the justification of the problem, then the main aims of the research, method and structure of the paper. The next section is on the literature review to present the research gap. In the case of your paper, you combine both parts into an introduction. Please divide the part on the work platform from the introduction, as actually starting from the second paragraph of the introduction should be classified as the literature review on the platform work. Properly re-phrase the introduction itself (which now in fact consists only of the first paragraph).

4.       In my mind, the explanation of how do the work platforms operate is needed in the paper.

5.       You wrote (p.1, lines 40-41): “However digital labour platforms might contribute to reinforcing traditional gender roles and relations, increasing gender inequality.” Here are my concerns:

5.1.    You write both about digital labour platforms (worrying about producing the labour segmentation by gender) and in the same paragraph about the platform work as “similar to (or a new form of) precarious work (MacDonald & Giazitzoglu, 2019; Huws et al., 2018; De Stefano, 2016), characterized by uncertainty, instability and insecurity, with workers bearing most of the risks and receiving limited protections (Kalleberg and Vallas, 2017), women who are disproportionately represented in precarious employment.” I feel confused with these sentences, as you point out the risk of sustaining gender segmentation due to the development of digital work platforms, and the same time you describe the platform work as precarious work, which is not the best work for anyone.

5.2.    I suppose (I’m rather guessing) that the confusion results from the different aspects of platform work. As far as you write on digital labour platforms, via which freelancers deliver their digital services to customers, the underrepresentation of women might be seen as a problem reinforcing the gender inequalities, but this underrepresentation is rather related to the female underrepresentation in digital works in general (problem of gender digital divide). However, as far as you later on focus on the food delivery platform work, via which cyclists deliver food ordered digitally, the situation of cyclists significantly differs from the situation of freelancers. In my opinion, to better express your research interests, you should present the different types of labour platforms, and refer to gender aspects in each type. Especially that you mentioned about: „digital labour platforms”, „delivery platforms”, „crowd-work platforms”.

6.       The general problem of gendered work is discussed, among others, in the dual labour market theory, assuming the existence of primary and secondary labour markets. By secondary labour markers’ characteristics, the following examples are meant: lower wages, lower promotion possibilities, lower social prestige, more difficult working conditions etc. From the gender perspective, women are overrepresented in the secondary labour markets. What kind of labour markets is the food delivery platform work? The description you started at p. 2, line 98, indicates that food delivery platform work is a physically demanding work, characterised by risk and limited safety, of rush nature, algorithmically managed etc. Such characteristic shows that it is rather an example of secondary labour market. Why do you suggest that such kind of jobs should have a higher representation of women? Please explain your point.

7.       The research method, a case study of a food delivery platform with in-depths interviews with founders and employees, is appreciated as a qualitative method.

8.       Truly speaking, I do not see any gendered-related aspects of the case study and interviews, but the business model of a company which is based on fair treatment of employees. Your results show (let me quote):

8.1.    “All riders have a regular contract, receive a wage not tied to the number of deliveries, and are covered by insurance, which is additional to the social protections and safeguards provided by the contract.”. I think all employees, not just women, prefer being employed full-time with fixed salaries.

8.2.    „the safe performance of work”. I think all employees, not just women, prefer to experience the safe performance of work.

8.3.    „no indications/impositions regarding the speed of deliveries, and the service is stopped in case of severe weather conditions.” I think all employees, not just women, prefer working without the speed delivery indication and not working during the severe weather condition, having still a fixed salaries.

8.4.    I might continue (safety, GPS tracking, resting place, clothing etc.), but I hope that my point is clear now. All arrangements implemented by founders are related to work conditions and, in my opinion, are welcomed by all employees, not just female employees. Please convince your potential readers that these work arrangements are expected by females, not by males to justify the gendered perspective of your research.

9.       I was also thinking, to which extent do you analyse the digital labour platform, or rather a company which hire and manage employees on the traditional manner (job contract with fixed salaries), but attracts clients through the digital platform. The answer on this question is important to realise to which extent does your paper answer the special issue call.

Author Response

Thank you for giving me the opportunity to submit a rewrite version of manuscript Women's tailored food delivery platform. The case of a small company in Italy.

We appreciate the time and effort that you and the reviewers dedicated to providing feedback on manuscript and I’m grateful for the insightful comments on and valuable improvements to my paper.

I have incorporated most of the suggestions made by the reviewers (Red changes in the uploaded file). Please see below, for a point-by-point response to the reviewers’ comments and concerns.

Q: The focus of the paper should be clear from the starting point. You refer to ‘food delivery platform’ in the title; to ‘platform work’ in the first sentence of the abstract; and to ‘women riders’ in the second sentence. I felt lost with your main topic after those first lines. Please rephrase these parts to be strict with your main issue from the beginning.

R: Thank you for this suggestion, I have reworded these parts to try and make them clearer.

Q: Abstract requires to be re-written. Look at your first and fourth lines of the abstract. In the first, you describe the aim (let me quote): “This paper aims to analyze the gender impact of platform work”, then you make some justification, and then again in the 4th line you come back to your aim, however, you write on different aim than in the first line: ”The aim is to understand how and in what ways (…)”. You need to be strict with describing your aim and with keeping the structure of an abstract: justification, aim(s), method, results, novelty.

R: I rewrote the abstract

Q: The introduction needs to be re-structured. Typically, an introduction presents the justification of the problem, then the main aims of the research, method and structure of the paper. The next section is on the literature review to present the research gap. In the case of your paper, you combine both parts into an introduction. Please divide the part on the work platform from the introduction, as actually starting from the second paragraph of the introduction should be classified as the literature review on the platform work. Properly re-phrase the introduction itself (which now in fact consists only of the first paragraph).

R: In the first version I had followed the journal's instructions to have a single paragraph between the introduction and the theoretical framework, now I have divided the two parts and rewritten the introduction. 

Q: In my mind, the explanation of how do the work platforms operate is needed in the paper.

R: I have included this specification in the introduction.

Q: You wrote (p.1, lines 40-41): “However digital labour platforms might contribute to reinforcing traditional gender roles and relations, increasing gender inequality.” Here are my concerns:

You write both about digital labour platforms (worrying about producing the labour segmentation by gender) and in the same paragraph about the platform work as “similar to (or a new form of) precarious work (MacDonald & Giazitzoglu, 2019; Huws et al., 2018; De Stefano, 2016), characterized by uncertainty, instability and insecurity, with workers bearing most of the risks and receiving limited protections (Kalleberg and Vallas, 2017), women who are disproportionately represented in precarious employment.” I feel confused with these sentences, as you point out the risk of sustaining gender segmentation due to the development of digital work platforms, and the same time you describe the platform work as precarious work, which is not the best work for anyone.

I suppose (I’m rather guessing) that the confusion results from the different aspects of platform work. As far as you write on digital labour platforms, via which freelancers deliver their digital services to customers, the underrepresentation of women might be seen as a problem reinforcing the gender inequalities, but this underrepresentation is rather related to the female underrepresentation in digital works in general (problem of gender digital divide). However, as far as you later on focus on the food delivery platform work, via which cyclists deliver food ordered digitally, the situation of cyclists significantly differs from the situation of freelancers. In my opinion, to better express your research interests, you should present the different types of labour platforms, and refer to gender aspects in each type. Especially that you mentioned about: „digital labour platforms”, „delivery platforms”, „crowd-work platforms”.

R: Thank you for this suggestion, I have rewritten the paragraph to make it clearer.

Q: The general problem of gendered work is discussed, among others, in the dual labour market theory, assuming the existence of primary and secondary labour markets. By secondary labour markers’ characteristics, the following examples are meant: lower wages, lower promotion possibilities, lower social prestige, more difficult working conditions etc. From the gender perspective, women are overrepresented in the secondary labour markets. What kind of labour markets is the food delivery platform work? The description you started at p. 2, line 98, indicates that food delivery platform work is a physically demanding work, characterised by risk and limited safety, of rush nature, algorithmically managed etc. Such characteristic shows that it is rather an example of secondary labour market. Why do you suggest that such kind of jobs should have a higher representation of women? Please explain your point.

R: Thank you for your suggestion, I agree with your objections and have tried to clarify my arguments further.

Q: The research method, a case study of a food delivery platform with in-depths interviews with founders and employees, is appreciated as a qualitative method.

Q: Truly speaking, I do not see any gendered-related aspects of the case study and interviews, but the business model of a company which is based on fair treatment of employees. Your results show (let me quote):

“All riders have a regular contract, receive a wage not tied to the number of deliveries, and are covered by insurance, which is additional to the social protections and safeguards provided by the contract.”. I think all employees, not just women, prefer being employed full-time with fixed salaries.

“the safe performance of work”. I think all employees, not just women, prefer to experience the safe performance of work.

“no indications/impositions regarding the speed of deliveries, and the service is stopped in case of severe weather conditions.” I think all employees, not just women, prefer working without the speed delivery indication and not working during the severe weather condition, having still a fixed salaries.

I might continue (safety, GPS tracking, resting place, clothing etc.), but I hope that my point is clear now. All arrangements implemented by founders are related to work conditions and, in my opinion, are welcomed by all employees, not just female employees. Please convince your potential readers that these work arrangements are expected by females, not by males to justify the gendered perspective of your research.

R: I explained more about the relevance of gender issues in the discussions.

Q: I was also thinking, to which extent do you analyse the digital labour platform, or rather a company which hire and manage employees on the traditional manner (job contract with fixed salaries), but attracts clients through the digital platform. The answer on this question is important to realise to which extent does your paper answer the special issue call.

R: Thank you for asking this question, but I believe that this is not simply a company using traditional employment methods, but rather a company that has decided to enter the world of digital platforms, but with different management and conditions precisely to develop women's employment in a booming industry. I have therefore made this clearer in the text.

Round 2

Reviewer 2 Report

Thank you for your clarifications and all adoptions in the paper.